# Impact of Wound Dressing Changes on Nursing Workload in an Intensive Care Unit

**DOI:** 10.3390/ijerph20075284

**Published:** 2023-03-28

**Authors:** Juliana Dias dos Reis, Pedro Sa-Couto, José Mateus, Carlos Jorge Simões, Alexandre Rodrigues, Pedro Sardo, João Lindo Simões

**Affiliations:** 1Centre for Research and Development in Mathematics and Applications (CIDMA), Department of Mathematics (DMAT), University of Aveiro, 3810-193 Aveiro, Portugal; julianadiasdr@gmail.com (J.D.d.R.); p.sa.couto@ua.pt (P.S.-C.); 2Intensive Care Unit, Centro Hospitalar do Baixo Vouga E.P.E., 3810-164 Aveiro, Portugal; zemateus@gmail.com (J.M.); carlos.simoes.12240@chbv.min-saude.pt (C.J.S.); 3School of Health Sciences (ESSUA), University of Aveiro, 3810-193 Aveiro, Portugal; alexandre.rodrigues@ua.pt (A.R.); pedro.sardo@ua.pt (P.S.); 4Centre for Innovative Biomedicine and Biotechnology (CIBB)—Center for Health Studies and Research, University of Coimbra, 3004-531 Coimbra, Portugal; 5Institute of Biomedicine (iBiMED), University of Aveiro, 3810-193 Aveiro, Portugal

**Keywords:** nursing, workload, intensive care units, wounds and injuries, TISS-28

## Abstract

The objective of this study is to understand how the type of wound dressing changes (routine or frequent) in patients admitted to intensive care units influences nurses’ workload. This study used a database of retrospective and analytical observational study from one Portuguese intensive care unit. The sample included 728 adult patients admitted between 2015 and 2019. The nursing workload was assessed by the TISS-28 scale, both at admission and at discharge. The linear regression results show that patients with frequent dressing changes are associated with a higher nursing workload, both at admission (Coef. 1.65; 95% CI [0.53; 2.77]) and discharge (Coef. 1.27; 95% CI [0.32; 2.22]). In addition, age influences the nursing workload; older people are associated with a higher nursing workload (at admission Coef. 0.07; 95% CI [0.04; 0.10]; at discharge Coef. 0.08; 95% CI [0.05; 0.10]). Additionally, an increase in nursing workload at admission would significantly increase the nursing workload at discharge (Coef. 0.27; 95% CI [0.21; 0.33]). The relative stability of the nursing workload over the studied years is also another important finding (the influence of studied years is non-significant). In conclusion, patients with frequent dressing changes presented higher TISS-28 scores when compared with patients with an exchange of routine dressings, which leads to a higher nursing workload.

## 1. Introduction

Nowadays, nursing workload assessment is widely discussed and implemented to qualify, plan, and evaluate intensive care units (ICU) [1,2]. Nursing workload is defined as a product of the average daily number of patients seen, adjusted by the degree of dependence and type of care and the average time of assistance for each patient [3].

An increase in nursing workload results in a reduced patient survival rate, which in turn may be attributable to the increased suboptimal care for some patients. As a result, it may affect the overall required care for some of the patients [4]. Low nurse staffing is associated with omissions of essential nursing care, identified as a key mechanism leading to adverse patient outcomes [5]. Hence, previous reports described the nurse-to-patient ratio to evaluate patient safety in relation to the nursing workload. Previous research exhibited that the nursing workload is a more complex correlation and cannot be determined by a simple ratio such as the nurse-to-patient one [4].

Nonetheless, published studies show that an intensification in working hours of the nursing team provided to specific patients is associated with a reduction in the occurrence of adverse events [6]. The nursing team spends approximately 70% focus on the treatment of only one single patient. Moreover, the nursing workload required by patients in intensive care was identified as a risk factor for the occurrence of adverse events, mostly derived from pressure ulcers and/or medication errors. Stays of longer than 3 days, a high acute physiology and chronic health evaluation II (APACHE II) score, coming from the surgery department, and having a diagnosis of trauma and emergency were associated with a high workload [6,7].

Evidence indicates that there are several factors related to the hospitalization of critically ill patients that potentially influence the nursing workload; however, studies on the influence of factors related to patients are rarer. Some of the variables referred to in the literature as potential influencers of nursing workload include gender, age, weight, length of stay, clinical status, adverse events, and patient death [1,8].

Several tools for assessing the nursing workload in intensive care have been presented. Therapeutic intervention scoring system 28 (TISS-28) has been the most used and recognized worldwide tool for measuring nursing workload in the context of critical patients [9,10].

In Portugal, the TISS-28 scale is an instrument broadly applied in ICUs, despite its weakness. The TISS-28 is classified as a trustworthy tool to assess the nurse’s workload, can be easily and rapidly applied with few resources, reflects the specificity of each patient in relation to severity, and allows a comparison of the workload between each patient or group of patients [1]. This scale is a system for measuring severity and nursing workload, which is based on the quantification of therapeutic interventions according to the complexity, degree of invasiveness, and time spent by nurses to perform certain procedures in critically ill patients. TISS-28 is composed of several evaluation categories: basic activities, ventilator support, cardiovascular support, renal support, neurologic support, metabolic support, and specific interventions. The 28 variables of TISS-28 are analyzed daily, allowing the achievement of a patient’s evolution profile by scoring and classifying the severity [11]. The use of this scale could therefore be advantageous in ICU planning, risk stratification, and resource allocation [12].

Wounds are a rising problem in hospitalized patients, especially in the ICU environment; consequently, they are related to the quality of care and are directly associated with increased length of hospital stay, risk of complications, and costs [13,14]. Accordingly, wound management of critical patients is a very important part of critical nursing care [15] and the type of wound dressing change is valued when assessing the nursing workload. Critical care nurses play a vital role in the early assessment and management of wound infection and in the detection of early signs of sepsis associated with this infectious focus [16].

Related to the wound’s treatment, in the TISS-28 category “basic activities”, the type of dressing change is recorded in two indicators: routine dressing changes (care and prevention of decubitus and daily dressing change) or frequent dressing changes (at least one time per each nursing shift) and/or extensive wound care [11]. Thus, the type of dressing performed by nurses is an important factor in the nursing workload in intensive care and can influence the overload of these professionals [17]. This issue is of extreme importance for a better understanding of the work performed by nursing teams and the knowledge of the factors that may be related to the workload of these professionals.

The present study aimed to understand how the type of wound dressing (routine dressing changes or frequent dressing changes) in patients admitted to intensive care units influences the nursing workload assessed by the TISS-28 scale.

## 2. Materials and Methods

### 2.1. Design

This is an observational, cross-sectional, and analytical study. This study was based on a retrospective analysis of electronic data recorded in PICIS^®^ (Critical Care Manager Software–version 8.2) used in an ICU of a hospital in the central region of Portugal from 1 January 2015 to 31 December 2019 (5 years).

### 2.2. Sample/Participants

The study population comprises critically ill adults and/or elderly patients to whom the TISS-28 scale was applied and who were admitted to one tertiary (level 3) ICU, implying a ratio of one nurse in the direct provision of care for every two patients hospitalized, 24 h a day.

The inclusion criteria were: (1) critically ill patients of both genders; (2) aged ≥ 18 years at the time of admission to the service; (3) inpatients admitted between 1 January 2015 and 31 December 2019; (4) patients with at least one assessment from the TISS-28 scale; (5) patients with at least one routine dressing change or frequent dressing changes in the first and last evaluation of the TISS-28. The exclusion criteria were: (1) critically ill patients with a length of stay of fewer than 24 h; (2) patients with an incomplete assessment of the TISS-28 scale.

According to the data available from the ICU, during the stipulated time of the study 740 critically ill patients were admitted to the ICU. Subsequent to the screening process, that is, after applying the previously described inclusion and exclusion criteria, a total sample of 728 adults were obtained (10 patients were excluded for being hospitalized for less than 24 h and thus did not have a TISS-28 assessment; 2 patients were also excluded because they did not have wound dressing changes at the time of the TISS-28 evaluation).

### 2.3. Data Collection

This study is retrospective and is based on the recording of sociodemographic, anthropometric, and clinical data and the scores of the TISS-28 Scale performed by ICU nurses. This scale is filled out at the beginning of the night shift, based on the care and procedures performed in the last 24 h.

The definition of the variables under study was limited to those variables to which we could have access to in the data files provided.

The independent variables gathered were:-Sociodemographic variables: gender; age; age categories (≤44 years, 45–64 years, 65–84 years, and ≥85 years); year of admission (2015; 2016; 2017; 2018; 2019);-Anthropometric variable: weight; weight categories (≤49 kg, 50–74 kg, 75–99 kg, and ≥100 kg);-Clinical variables: length of stay, presented in days (LOS); categorized length of stay (1 day, 2–7 days, 8–14 days, and ≥15 days); type of wound dressing (routine dressing changes and frequent dressing changes).


The outcome variables defined were:
-TISS-28 (first): the results of the first measurement 24 h after admission;-TISS-28 (last): TISS-28 measured at discharge;-Categorized TISS-28 assessment (Cullen Classes): Class I (up to 9 points), Class II (from 10 to 19 points), Class III (20 to 39 points), and Class IV (above 39 points).

The categories of the variable “type of wound dressing” were obtained through the following items of the TISS-28 scale:-Exchange of routine dressings: care and prevention of decubitus ulcers and daily dressing change (item 5 of the category “Basic Activities” of the TISS-28 scale);-Frequent dressing changes: frequent dressing change (at least once per nursing shift) and/or extensive wound care (item 6 of the category “Basic Activities” of the TISS-28 scale).

### 2.4. Validity and Reliability/Rigor

The TISS-28 scale is a system to measure the severity and workload of intensive care nursing based on the quantification of the interventions performed on inpatients in ICUs [18] (see Appendix A Table A1). Additionally, Miranda et al. [19] translated and validated this system into European Portuguese, enabling its use in the context of intensive care in Portugal. In this study, the reliability of data collection was high, with intraclass correlation coefficients greater than 0.90.

The measurement is daily by the procedures performed on the patient and, as a result, a single TISS-28 point corresponds to 10.6 min of the time of a nurse in direct care [9]. Depending on the total number of points obtained, the patients are classified into four groups according to the need for surveillance and intensive care (Cullen Classes):-Class I: patients who do not need to be in an intensive care service;-Class II: patients with an indication for admission to an intensive care service;-Class III: patients who require intensive care due to hemodynamic instability;-Class IV: patients who have a compulsive indication for the use of intensive care because they have great hemodynamic instability [18].

### 2.5. Ethical Considerations

The study was authorized by the hospital council board and ethics committee approval (File No. 32/01/2020). Confidentiality of the data and anonymity of the participants in the study were always guaranteed, as well as their treatment with respect and professional secrecy. Each patient was coded numerically in chronological order, making it possible to guarantee and maintain the anonymity of the participants throughout the study.

### 2.6. Data Analysis

For the analysis of sociodemographic and clinical data, descriptive statistics were used: absolute frequencies (n) and percentages (%); means (M) and standard deviations (SD).

The sociodemographic and clinical characterization of the patients was carried out considering the type of wound dressing (routine dressing changes and frequent dressing changes); then, a statistical test was performed to infer if there were statistically significant association/differences between them. Between two categorical variables, the statistical test used was the chi-squared and, for comparison between the two independent groups, the statistical test used was the Mann–Whitney test. Subsequently, a correlation analysis (Spearman rank test) was performed between the variables TISS-28 (first) and TISS-28 (last) per year and per type of wound dressing.

Multiple linear regressions were performed to identify which independent variables could be considered as predictors for the outcome variables: TISS-28 (first) and TISS-28 (last). Several statistical models were tested and analyzed. For Model 1, only the variables “type of wound dressing” (routine dressings as the reference group; frequent dressing), the “year” (2015 as reference group; 2016; 2017; 2018; 2019), and the interaction between the two variables were considered (defined as the full factorial model). In Model 2, as the interaction was not significant, only the two previous variables were considered (defined as the main effects model). In Model 3, in addition to these two variables, the following covariates were considered: “gender” (male was the reference group; female), “age”, “weight”, and “LOS” (only for the outcome TISS-28 (last)). Finally, Model 4 was only calculated for the variables “type of wound dressing” and “year”, with the covariates that had a *p*-value of less than 0.20 [20]. For TISS-28 (last), a model incorporating the influence of TISS-28 (first) was also presented (defined as TISS-28 (last)-VB model). The results of Model 4 are presented in the next section, while the results of Models 1, 2, and 3 are presented in Appendix C.

All results where *p*-value < 0.05 were considered significant. All statistical analysis was performed using the R software version 4.2.1 and the following packages: “epiDisplay”, “dplyr”, “ggpurb”, “ggplot2”, “tidyr”, “ploty”, “carData”, “lsmeans”, “emmeans”, and “FSA”.

## 3. Results

### 3.1. Sociodemographic and Clinical Characteristics of Participants

Table 1 shows the results related to the characterization of the 728 critically ill patients who took part in this study, considering the type of wound dressing present in the assessment, routine dressing changes (RDC), or frequent dressing changes (FDC).

In relation to the type of wound dressing changes, 73.9% of the participants had RDC and 26.1% had FDC. Regarding the gender, most participants were male, regardless of the type of wound dressing changes (61.1% for RDC and 57.8% for FDC). As for age, the most represented category is between 65 and 84 years old in both groups (44.6% for RDC and 52.6% for FDC). Regarding weight, in the group of patients with RDC, the most represented categories, with equal percentages, are 50–74 kg and 75–99 kg (43.1%). Regarding the group of patients with FDC, the most represented category was that of 50–74 kg (44.3%). Most patients were hospitalized between 2 and 7 days in both groups (52.2% for RDC and 54.2% for FDC).

According to the comparison of the groups, significant differences were observed in the LOS categ. variables (*p*-value = 0.047), regarding the days of stay in the intensive care unit, in this case the categorized variable.

Additionally, significant differences were obtained in the continuous variables under study, namely age, weight, and LOS (*p*-value < 0.001), and for the variable age and weight, the values are higher in the FDC group.

Regarding TISS-28, at admission and discharge there are statistically significant differences between the RDC and FDC groups (*p*-value < 0.001, both analyses), with higher TISS-28 values in the FDC group. Moreover, regarding the Cullen Classes variable, in the RDC group and the FDC group, despite more than half of the patients being in Class III (78.3% and 68.2%, respectively), the percentage of the FDC group in Class IV is 29.7 compared with 19.1% for the RDC group (*p* = 0.006).

### 3.2. TISS-28 (First) and TISS-28 (Last) by Year and by Type of Wound Dressing Changes

In Figure 1, it is possible to observe the distribution of TISS-28 at admission values (TISS-28 (first)) and TISS-28 at hospital discharge values (TISS-28 (last)) by year, clustered by type of wound dressing changes. It is possible to observe that most of the boxplots presented are overlapping over the years between RDC and FDC groups. Note that the values for TISS-28 (first) and TISS-28 (last) are very stable throughout the analyzed period. In Appendix B Table A2, more detailed data can be seen about the behavior of the variables studied by year.

### 3.3. Correlation Analysis Regarding TISS-28 (First) and TISS-28 (Last) by Year and by Type of Wound Dressing Changes

A correlation analysis was performed between the variable TISS-28 (first) and TISS-28 (last) per year; the *p*-values were significant for all years and the dependence was positive for all years, with coefficient values between 0.272 and 0.434. The values obtained by applying Pearson’s correlation do not differ from the values obtained with Spearman’s correlation. Therefore, it is possible to infer that the TISS-28 value at discharge is influenced by the TISS-28 at admission (see Table 2 for TISS-28 (last)-VB results) and the higher the value at admission, the higher the TISS-28 value at discharge will be. When studied by type of wound dressing changes, the values found for the correlation showed the same pattern previously explained.

### 3.4. Multiple Linear Regression Analysis Results Regarding TISS-28 (First) and TISS-28 (Last)

Two regressions were performed to analyze the relationship between the independent variables, namely “type of wound dressing changes”, “year”, “age”, “weight”, “LOS” and the outcome variable “TISS-28 (first)” or “TISS-28 (last)”. In the regression relating to TISS-28 (first), the LOS variable was not included in the model, as TISS-28 (first) relates to the first analysis performed within the first 24 h of hospitalization.

The results of the multiple linear regression Model 4 (wound dressing changes + year + covariates with *p*-value <0.20) for the TISS-28 in the first and last assessment (TISS-28 (first) and TISS-28 (last)-VA, respectively), can be observed in Table 3. The influence of TISS-28 (first) on TISS-28 (last) is also presented (see TISS-28 (last)-VB column). The results of other linear regression models can be observed in Table A3 (Appendix C).

For TISS-28 (first), the FDC were more likely to present higher values in the nursing workload at admission. Concerning the “age” variable, a relation was found with TISS-28 (first), so the increase in age is directly proportional to the increase in the nursing workload, with a greater possibility of older patients presenting higher TISS-28 values.

Regarding the linear regression for TISS-28 (last)-VA, it was found that patients with FDC are associated with higher values of TISS-28 (last). Regarding the age variable, a relation was also found with TISS-28 (last) and the increase in age is directly proportional to the increase in the nursing workload.

For TISS-28 (last)-VB, it was found that the TISS-28 (first) assessment had a significant impact on the results, while the variable “type of dressing changes” was non-significant. An increase in TISS-28 (first) would also increase the TISS-28 (last). Age remained significant, as in the previous models. If we considered “year” as a continuous variable in the models, this variable would remain non-significant and the coefficients for the significant variables would remain almost the same (results not presented).

## 4. Discussion

Being admitted to a hospital is an important measure of frailty [21], whereby the patients admitted to intensive care units [22] are normally associated to higher levels of nursing workload [1,2].

Nursing activities in the ICU are variable based on many factors such as working atmosphere, disease severity, workload, personnel qualifications and skills, and cost-efficacy, together with the determined clinical outcomes of the patients [23]. Therefore, providing adequate nursing staffing has become an essential component in preserving and providing better care quality and is directly associated with better patient satisfaction and improved clinical outcomes.

ICU nurses are constantly persuaded by various causes of stress in the work environment, namely fatiguing workload, reduced staffing, and complexity of procedures. Due to the exhaustive and stressful work, nurses are more likely to develop occupational stress, which is an important determinant of depression and burnout [24]. Regarding nursing workload, there is a high rate of burnout among professionals working in Portuguese ICUs, with 31% having a high level of burnout. Higher levels of burnout are associated with conflicts, ethical decision making regarding withdrawing treatments, and having a temporary work contract [25].

According to the study results, the nursing workload assessed through TISS-28 was identical over the 5 years under analysis, which reflects the high complexity of the patients who were admitted in this unit in the study period. Concerning the correlation analysis, it was possible to see that the higher TISS-28 (first) scores at admission, the higher the TISS-28 (last) scores at discharge. This reveals that patients who had a greater clinical complexity (and greater need for healthcare) upon admission to the UCI maintained that same need in the last evaluation carried out. It was not possible to understand whether the nursing workload remained high in the last assessment, associated with the maintenance or worsening of the patient’s clinical status, because, as this is a retrospective study, these specific data were not available.

Analyzing the international literature about the assessment of nursing workload in intensive care units, it was found that most of the studies use the TISS-28 [9], the nine equivalents of nursing manpower use score (NEMS) [19], and/or the nursing activities score (NAS) [26,27]. As previously mentioned, in Portugal the most used scale in intensive care units is the TISS-28, despite its limitations in assessing the total workload of nurses once that TISS-28 only covers about 43.3% of nursing activities [27].

This is one of the first studies to analyze the characteristics of the participants with “routine dressing changes” and “frequent dressing changes” (two items of the category “Basic Activities” of the TISS-28 scale) and to correlate them with different variables and with nursing workload.

These results showed that the participants with “frequent dressing changes” presented higher TISS-28 scores when compared with the participants with “routine dressings changes”. These data were particularly important in older and heavier participants with longer lengths of intensive care unit stays, since these are relevant comorbidities to wound development and have a great influence in the healing [28].

The frequency of dressing change is associated with the wound characteristics and evolution, type of dressing, and clinical compliance or protocol-specified change frequency, which is determinant to define the workload impact [29].

In the regression model, it was possible to observe that for TISS-28 at admission and at discharge, the significant variables for the model are: type of wound dressing changes and age. Hence, patients with frequent changes of wound dressings and older patients have a greater nursing workload.

The assessment of care needs and nursing workload assumes a prominent role [30] with the intention to reconcile care quality, resource optimization, and cost reductions [31,32]. Health management becomes a higher challenge when patients present simple and/or complex wounds [33,34,35]. It is very important to reinforce the idea that it is better to work on wound prevention (low-cost interventions and resources) than on wound treatment (complex interventions and expensive resources), aiming to reduce the health financial costs and the intangible costs to the patient, to the professional, and to the family caregiver [36].

Subsequently, it is essential to identify objective indicators that measure the real needs of the patients and of the nursing professionals, specifically in patients with wounds and injuries of different aetiologias, to ensure that all patients are offered consistently high-quality care in intensive care units [2,37].

In this sample, most of the participants required intensive care due to hemodynamic instability (Cullen Class III), which also influenced their care needs and, consequentially, the nursing workload. Concerning this result, it is important to highlight that when the patients had a higher level of instability, the development risk was higher in these patients and they could need more wound prevention interventions [38].

Furthermore, due to the data collection strategy used in this study, it was not possible to characterize each wound. However, the data showed that all the study participants needed at least basic care activities related to the care and prevention of pressure ulcers and daily dressing changes, while 26.1% needed frequent dressing changes (at least once per nursing shift) and/or extensive wound care. These data were consistent with other studies developed in intensive care units that highlight the need to preferentially assess the nursing workload that each participant generates [39] and the specificities of each shift [40,41] instead of calculating nursing ratios according to occupancy rates, average nursing workload per participant, and/or the total 24-h nursing workload in the intensive care unit [2]. Specifically, regarding the wound treatment, when the wound is complex, more time is needed, as well as more differentiated interventions by different health professionals aiming to choose the best dressing options [42].

Additionally, attention to the typology of each unit and patients’ (clinical) characteristics should be considered [43]. The (new) epidemiological challenges with critical care patients [2] raise the awareness of health professionals and institutions for this issue by creating specific training and research strategies [44,45]. Thus, in addition to the critical care needs of patients, the number of dressings, their complexity, and the nursing workload they represent can be considered to reduce the ratio of patients per nurse (one patient per one nurse, for example).

Nursing care is a critical point in the outcomes of hospitalized patients with wounds, but these outcomes are also influenced by the severity and complexity of the patients’ conditions. Nurses have underlined the importance of assessing wounds and dressings, aiming to continuously pursue the development of protocols for registration and proper treatment [46]. Intensive wound care can be achieved using trained critical care nurses who widely understand each patient’s physiological condition and wound severity [22].

Strategies related to wound care involve a meticulous assessment of the patient, risk stratification, and implementation of preventive measures aimed specifically at the patient’s conditions. Preventing wound development is relevant to avoiding prolonged lengths of hospital stays (and therefore infection) and managing costs to the public health sector and great suffering for patients and family members [47].

Nogueira et al. [17] identified that the addition of an affected body region increased the chance of the patient requiring a higher nursing workload by 33%. This study concluded that the increase in the number of affected body regions, together with the patient’s physiological severity, increases the time spent performing hygiene procedures, dressing changes, monitoring and titration, as well as increases the number of nursing professionals required to move and position the patient, thus generating high workload scores.

The fact that this is a retrospective study, based on information recorded in a health information system used in the intensive care unit, represents a limitation to this study as it was not possible to access some data/variables that would allow a better understanding of the patients’ profiles and the characteristics of the wounds they had. Thus, we are aware that this clinical importance should be not ignored and the workload is not simplified with the number of dressings.

Additionally, the score assigned to the two basic activities under analysis (routine dressing changes or frequent dressing changes) was the same (1 point) and this may have influenced the relevance of the differences found in the final TISS-28 score for each variable under analysis. Hence, the significant results that were found in this study regarding the final score of the TISS-28 are probably not merely influenced by the basic activities of changing wound dressings but also by the other items that the scale evaluates.

## 5. Conclusions

The presence of frequent dressing changes and/or extensive wound care is a challenge in ICU settings, not only for clinical reasons but also for the nursing workload that is raised.

In this study, a relative stability of the nursing workload was found over the studied years for both types of wound dressing changes. It was likewise concluded that patients with frequent dressing changes were associated with a higher nursing workload, both at admission and discharge. Similarly, higher ages influenced the nursing workload, increasing it at admission and at discharge.

These results allowed to understand that performing frequent dressing changes is a timely consuming intervention in intensive care units, influencing the management of human resources in nursing concerning the number of patients per nurse during work shifts.

Therefore, we hope that this study can contribute to the implementation of measures to improve performance in nursing interventions, contributing to better resource management, redefinition of care priorities, reduction of workload, burnout symptoms, and associated adverse events, and reduction of additional costs, among others.

## Figures and Tables

**Figure 1 ijerph-20-05284-f001:**
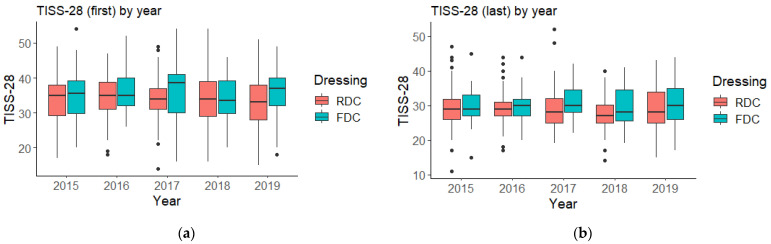
TISS-28 (first) (**a**) and TISS-28 (last) (**b**) by year, cluster by type of wound dressing changes (routine dressing changes (RDC) vs. frequent dressing changes (FDC)).

**Table 1 ijerph-20-05284-t001:** Sociodemographic, anthropometric, and clinical characterization of critically ill patients during the period 2015–2019 (*n* = 728) by type of dressing changes.

	Type of Dressing Changes			
	Routine Dressing Changes (*n* = 538)	Frequent Dressing Changes (*n* = 190)		
**Categorical Variables**	*n* (%)	*n* (%)	Statistic Result	*p*-Value
Gender				
Male	330 (61.1%)	111 (57.8%)	χ^2^ (1) = 0.39	0.535
Female	208 (38.5%)	79 (41.1%)		
Age categ. (years)				
≤44	85 (15.7%)	28 (14.6%)	χ^2^ (3) = 4.55	0.208
45–64	184 (34.1%)	51 (26.6%)		
65–84	241 (44.6%)	101 (52.6%)		
≥85	28 (5.2%)	10 (1.0%)		
Weight categ. (Kg)				
≤49	10 (3.2%)	6 (4.9%)	χ^2^ (3) = 1.87	0.599
50–74	135 (43.1%)	54 (44.3%)		
75–99	135 (43.1%)	46 (37.7%)		
100	33 (10.5%)	16 (13.11%)		
Missing	227 (42.0%)	70 (36.5%)		
LOS categ. (days)				
1	55 (10.2%)	31 (16.1%)	χ^2^ (3) = 7.93	0.047
2–7	282 (52.2%)	104 (54.2%)		
8–14	118 (21.9%)	29 (15.1%)		
≥15	83 (15.4%)	26 (13.5%)		
**Continuous variables**	M ± SD	M ± SD	Statistic result	*p*-value
Age (years)	62.18 ± 16.51	64.03 ± 16.79	*U* = 531440	<0.001
Weight (Kg)	77.94 ± 18.09	78.28 ± 19.25	*U* = 316680	<0.001
LOS (days)	7.94 ± 10.51	7.04 ± 7.22	*U* = 482197	<0.001
**Outcome variables**	M ± SD	M ± SD	Statistic result	*p*-value
TISS-28 (first)	33.72 ± 6.72	35.47 ± 7.23	*U* = 531440	<0.001
TISS-28 (last)	29.89 ± 5.62	30.24 ± 5.51	*U* = 109310	<0.001
TISS-28 (categ.)	*n* (%)	*n* (%)	Statistic result	*p*-value
Cullen Classes				
Class I	0 (0.0%)	0 (0.0%)	χ^2^ (2) = 10.269	0.006
Class II	12 (2.2%)	2 (1.0%)		
Class III	423 (78.3%)	131 (68.2%)		
Class IV	103 (19.1%)	57 (29.7%)		

**Table 2 ijerph-20-05284-t002:** Correlation analysis between TISS-28 (first) and TISS-28 (last) by year and by type of wound dressing changes.

Year	Total (Coef.)	Routine Dressing Changes at Admission and Discharge (Coef.)	Frequent Dressing Changes at Admission and Discharge (Coef.)
2015	0.302 *** (*n* = 153)	0.271 ** (*n* = 98)	0.211 (*n* = 17)
2016	0.272 ** (*n* = 103)	0.263 * (*n* = 64)	0.476 (*n* = 10)
2017	0.323 *** (*n* = 104)	0.290 * (*n* = 60)	0.199 (*n* = 17)
2018	0.301 *** (*n* = 124)	0.372 *** (*n* = 76)	0.224 (*n* = 19)
2019	0.434 *** (*n* = 159)	0.503 *** (*n* = 98)	0.426 * (*n* = 31)
2015–2019	0.338 *** (*n* = 643)	0.355 *** (*n* = 396)	0.320 * (*n* = 94)

* *p* < 0.05; ** *p* < 0.01; *** *p* < 0.001.

**Table 3 ijerph-20-05284-t003:** Multiple linear regression analysis results for the final model for TISS-28 (first) and TISS28 (last)-VA and TISS28 (last)-VB. In bold are presented the significant results.

Variables	TISS-28 (First) (*n* = 728)		TISS-28 (Last)-VA (*n* = 633)		TISS-28 (Last)-VB (*n* = 633)	
	Coef.	95%CI	Coef.	95%CI	Coef.	95%CI
Intercept	**29.3**	**[27.1; 31.4]**	**24.1**	**[22.3; 26.0]**	**16.2**	**[13.6; 18.7]**
Wound Dressing Changes (Ref. group: routine dressing changes (RDC))						
Frequent Dressing Changes (FDC)	**1.65**	**[0.53; 2.77]**	**1.27**	**[0.32; 2.22]**	0.55	[−0.36; 1.47]
Year (Ref. Group: 2015)						
2016	0.56	[−1.02; 2.13]	0.03	[−1.33; 1.40]	−0.07	[−1.37; 1.22]
2017	0.47	[−1.12; 2.05]	0.62	[−0.75; 2.00]	0.54	[−0.80; 1.84]
2018	−0.20	[−1.73; 1.32]	−1.16	[−2.46; 0.13]	−1.08	[−2.30; 0.15]
2019	−0.58	[−2.01; 0.84]	0.29	[−0.93; 1.51]	0.42	[−0.72; 1.57]
Age	**0.07**	**[0.04; 0.10]**	**0.08**	**[0.05; 0.10]**	**0.06**	**[0.03; 0.08]**
TISS-28 (first)	n.a.	-	n.a.	-	**0.27**	**[0.21; 0.33]**

n.a.—not applicable; TISS-28 (First): R^2^ = 0.04; TISS-28 (Last)-VA: R^2^ = 0.07; TISS-28 (Last)-VB: R^2^ = 0.15.

## Data Availability

The data presented in this study are available on request from the corresponding author. The data are not publicly available due to privacy and ethical restrictions.

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
