# Peer review of "Impact of Wound Dressing Changes on Nursing Workload in an Intensive Care Unit"

_ijerph, 2023, doi:10.3390/ijerph20075284_

Round 1

Reviewer 1 Report

I would like to thank the authors for their original titles and detailed studies and their contributions to science. My suggestion for the text is as follows:

1. In this study, the authors wanted to reveal the reflection of routine and frequent wound dressing work in intensive care units as a nursing workload, according to numerical data. As they honestly stated, they did not share any data about the type of wound, the need for frequent dressing, and the medical condition of the patient, apart from numerical data. In order for patient care to be effective, attention should be paid to the messages to be given in the research so that this clinical importance is not ignored and the workload is not simplified with the number of dressings. In both the introduction and the conclusion section, the authors should pay attention to the messages they will give.

2. In addition, the number of nurses per patient is not specified in the article. One of the aims of the article is to reveal the workload for quality nursing care. Giving suggestions on how the workload should be managed instead of just determining the workload (for example: the number of nurses per patient with frequent wound care) may lead to a more significant contribution of the authors to the literature.

3. It would be appropriate to use the word "gender" instead of the word "sex" when describing gender.

Author Response

Manuscript ID: ijerph-2222497entitled "IMPACT OF ROUTINE OR FREQUENT WOUND DRESSING CHANGES ON NURSING WORKLOAD IN AN INTENSIVE CARE UNIT: A 5-YEAR RETROSPECTIVE ANALYSIS",

Dear Reviewer,

Thank you for allowing us the opportunity to revise this manuscript.

Editor and reviewers’ comments have been extremely useful.

We have provided a point-by-point response to each suggestion in the attached table.

The paper has been restructured to provide greater clarity.

We look forward to your response and decision in due course.

Yours Sincerely,

Reviewer 2 Report

Thank you for providing me with the opportunity for reviewing this interesting manuscript. I have only a few minor comments that must be taken into account, they may improve the quality of the manuscript.

In the Introduction need to highlight on nursing role in wound and sepsis and important of patient safety and how these concept affect ICU nurses work.  Kindly use this articles. 

·         Nurses' knowledge, attitudes, practice, and decision-making skills related to sepsis assessment and management.

·        

Discussion and conclusion

Briefly comment on why the study is necessary without repeating the sentences used in the introduction, and include international studies. Kindly use this articles. 

Summary of key findings (primary outcome measures, secondary outcome measures, results as they relate to a prior hypothesis); compare with findings from other studies.

Author Response

(The authors gave the same response as above.)

Reviewer 3 Report

Thank you so much for the opportunity to review this interesting study. Overall, the study and its findings are very interesting. However, the authors do not adhere to the authors' instructions of the International Journal of Environmental Research and Public Health. Thus, major revisions are required. Following are some comments:

·      Even though the title reflects the aim and methodology of the study, the title is very long and could be shortened.

·      The abstract is more than 300 words, and the authors must reduce the number to 200 words, as indicated in the authors' instructions.

·      The abstract lacks a summary of the article's main findings (i.e., the article lacks numbers and statistics).

·      The introduction of the manuscript needs major revision with a focus on the nursing workload in intensive units, a summary of international perspectives on the topic, and an analysis of the needs for conducting the study.

·      The background is very long and is not required by the International Journal of Environmental Research and Public Health; the authors must shorten and incorporate it into the introduction.

·      The methods are very long and must be rewritten concisely

·      The reference list is written in different styles. As required by the International Journal of Environmental Research and Public Health, references must be numbered in their order of appearance in the text (including table captions and figure legends) and listed individually at the end of the manuscript.

Author Response

(The authors gave the same response as above.)

Round 2

Reviewer 2 Report

None 

Reviewer 3 Report

My pleasure to read the revised manuscript.  All comments are addressed sufficiently. Therefore, I recommend the manuscript for publication. Many thanks.